# On the Use of Pseudo-Protic Ionic Liquids to Extract Gold(III) from HCl Solutions

**DOI:** 10.3390/ijms24076305

**Published:** 2023-03-27

**Authors:** Francisco Jose Alguacil, Jose Ignacio Robla

**Affiliations:** Centro Nacional de Investigaciones Metalurgicas (CSIC), Avda. Gregorio del Amo 8, 28040 Madrid, Spain

**Keywords:** gold(III), hydrochloric acid, pseudo-protic ionic liquids, extraction, stripping, zero gold valent, nanoparticles

## Abstract

Solvent extraction of gold(III) from HCl media using pseudo-protic ionic liquids (PPILs) dissolved in toluene as the extractant phase is investigated. Three PPILs are generated from the reaction of commercially available amines and 1 M HCl solution and named as pri-NH_2_H^+^Cl^−^ (derived from the primary amine Primene 81R), sec-NHH^+^Cl^−^ (derived from the secondary amine Amberlite LA2) and ter-NH^+^Cl^−^ (derived from the tertiary amine Hostarex A327). In the above structures, -NH_2_H^+^Cl^−^, -NHH^+^Cl^−^ and -NH^+^Cl^−^ represented the active groups (anion exchangers) of the respective PPIL. In the case of gold(III) extraction, the experimental variables investigated included the equilibration time (2.5–30 min), temperature (20–60 °C), HCl concentrations (1–10 M) in the aqueous phase, gold(III) concentration (0.005–0.05 g/L) in this same phase, and PPILs concentrations in the organic phase. From the experimental data, and using the Specific Interaction Theory, the interaction coefficients (ε) for the pair AuCl_4_^−^, H^+^ are estimated for the systems involving the three PPILs. Gold(III) is recovered from the metal-loaded organic phases using sodium thiocyanate solutions, and from these, gold is finally recovered by the precipitation of zero-valent gold (ZVG) nanoparticles.

## 1. Introduction

In these first years of the third decade of 21st century, and similarly to past centuries, gold is one of the metals, if not the metal, that attracted a major interest from humans. This is due to its inherent attractiveness and rarity, but also to the multiple applications in which this precious metal takes part, which makes its price always high (about USD 1900 per ounce at the time of writing this article (mid-February 2023)).

Gold is recovered from its raw materials by bioleaching or cyanidation, followed by other operations, i.e., adsorption on the activated carbon [1,2,3], zinc precipitation [3,4] and ion exchange [2,5]. However, in recent years, and due to environmental and economic issues, the recovery of this metal (and others) from the wastes generated by humans is of increasing interest, and this is how the concepts of the circular economy and urban mining come in modern life.

Into the concept of urban mining, gold appeared in a series of solid wastes, such as electronic devices and scraps, printed circuit boards, jewelry scraps, etc., and generally after the separation of undesirable components, gold is first dissolved from the solid waste via acidic leaching in aqua regia medium, though the use of ionic liquids as the leaching medium is also considered [6]. From this highly acidic medium, gold is separated from other less valuable metals by different separation technologies prior to its final recovery step.

Considering these various separation technologies, recent published applications of these on gold(III) processing included solvent extraction [7,8,9,10,11], precipitation [12,13], ion exchange [14,15,16], adsorption [17,18,19,20,21] and even polymer inclusion membranes [22].

In recent years, and especially in the case of solvent extraction operations, a group of chemicals under the label of green solvents, such as ionic liquids, is receiving interest for being used as potential extractants for many metals. This importance is due to some of their properties, such as low vapor pressure, low volatility, low flammability, thermal stability, etc. [23], though their friendly environmental status is under debate [24,25]. Ionic liquids can also be divided into some subfamilies [26]: (i) aprotic ionic liquids and task-specific ionic liquids, both types considered as fully ionic; (ii) protic ionic liquids, generated by the reaction of an acid and a base; and (iii) pseudo-protic ionic liquids (PPILs), which are generated by the reaction of an acid and a base, but sometimes considered not fully ionic. A special case of these (iii) compounds is the PPILs formed from tertiary amines [27,28], and also the PPILs derived from primary and secondary amines.

The present work investigates the use of the solvent extraction operation for the recovery of gold(III) from HCl solutions; as extractants for the precious metal, three PPILs dissolved in toluene, generated from the reaction of a primary, secondary and tertiary amine with HCl, are considered. The use of a diluent in the organic phases, such as toluene, which fulfills the security regulations, is considered a necessity because it (i) decreased the high viscosity of the PPILs, favoring phase separation and organic phase flow in the case of scaling up the operation to a continuous mode, and (ii) allowed an adequate range of extractant concentrations for each particular use, avoiding maintaining unused extractant (probably the most expensive item of the inventory) in the solvent extraction circuit. Gold(III) extraction is investigated under various experimental variables, and the experimental data are used to estimate the values of the AuCl_4_^−^,H^+^ pair interaction coefficient for each of the three extraction systems. After stripping, gold is precipitated from the stripped solution as zero-valent gold (ZVG) nanoparticles by the use of sodium borohydride.

## 2. Results and Discussion

### 2.1. Generation of the PPILs

It is known that amines are bases, thus they react with mineral acids to form the correspondent salt accordingly with the general reaction:(1)$$R_{3}N_{\mathrm{org}}+H_{\mathrm{aq}}^{+}+X_{\mathrm{aq}}^{-}\Leftrightarrow R_{3}NH^{+}X_{\mathrm{org}}^{-}$$

In the above equation, the subscripts org and aq refer to the organic and aqueous phases, respectively, and the equilibrium is shifted to the right depending on the amine basicity. The species formed in the organic phase is a quaternary ammonium salt, which in other terms is a pseudo-protic ionic liquid. The same type of equilibria result from the reaction of the primary and secondary amines with mineral acids.

In the present work, the PPILs used in the investigation are generated from the reaction of the primary amine Primene 81R, the secondary amine Amberlite LA2 or the tertiary amine Hostarex A327, dissolved in a toluene medium with 1 M HCl solution. The experiments are carried out using equal volumes of both phases, equilibrated during 5 min at 20 °C. The results from these experiments are summarized in Table 1. In all the cases, the rate of conversion from the amine to the PPIL exceeds 98%.

The experimental results are numerically treated by a tailored computer program that minimizes the U function, defined as:(2)$$U=\Sigma {\left(\log D_{\exp}-\log D_{\mathrm{cal}}\right)}^{2}$$
where D_exp_ and D_cal_ are the experimental and calculated-by-the-program distribution coefficients (D_HCl_), respectively. The values of the respective log K_ext_, for the reactions as in Equation (1), are also given in Table 1.

### 2.2. Gold(III) Extraction by the PPILs

#### 2.2.1. Influence of the Equilibration Time on Gold Extraction

The influence of this variable on gold(III) extraction was investigated using aqueous solutions containing 0.01 g/L Au(III) in 2 M HCl medium and the organic phases of the corresponding PPIL (2.1 × 10^−2^ M in toluene). The experiments were carried out at 20 °C, using a V_org_/V_aq_ relationship of 1 and equilibration times from 2.5 to 60 min.

The results showed that in all three cases, equilibrium was reached within 5 min of the contact time between both phases, and with extraction rates of 82% in the case of priNH_2_H^+^Cl^−^ and in excess of 99% for secNHH^+^Cl^−^ and terNH^+^Cl^−^, which demonstrated that gold(III) extraction seemed to be more favorable in the case of these two PPILs than in the case of the reagent derived from the primary amine.

#### 2.2.2. Influence of the Temperature of Gold Extraction

The variation of gold extraction at various temperatures (20–60 °C) using the PPILs was also investigated using the same aqueous solution from above and organic phases of the corresponding PPIL in toluene (Table 2).

The results derived from these set of experiments are shown in Table 1. It can be seen that in the 3 cases, gold extraction decreased with the increase of the temperature from 20 °C to 60 °C; thus, the gold distribution coefficient values (D_Au_) decreased with the increase of the variable. At a first approximation:(3)$$\log D_{\mathrm{Au}}=\frac{\Delta S^{0}}{2.3R}-\frac{\Delta H^{0}}{2.3R}\frac{1}{T}$$
and a plot of log D_Au_ versus 1000/T allowed estimating both ΔS° and ΔH° for the three extraction systems. These values are shown in Table 2, indicating the exothermic nature of the gold extraction process and a decrease in randomness with the loading of the precious metal in the respective organic phase.

Since:(4)$$\Delta G^{0}=\Delta H^{0}-T\Delta S^{0}$$
the values of ΔG° for each of the extraction systems are given also in Table 2. All the systems were spontaneous.

#### 2.2.3. Influence of the PPIL Concentration in the Organic Phase on Gold Extraction

To investigate the influence of this variable on gold extraction, aqueous phases containing 0.01 g/L Au(III) in different HCl medium (1–10 M) were put into contact with organic solutions containing different PPILs concentrations in toluene. The results from these set of experiments are shown in Figure 1 for the PPILs priNH_2_H^+^Cl^−^, secNHH^+^Cl^−^ and terNH^+^Cl^−^.

Comparison of the three figures showed that a general pattern occurred in the three systems: (i) there was a logical increase of the percentage of gold extraction with the increase of the extractant concentration in the organic phase, (ii) there was a maximum of gold extraction for HCl concentrations in the 2–5 M concentration range, and (iii) a slight decrease in the extraction occurred with the 6 M HCl concentration, though this decrease was only observed with the lower extractant concentrations 2.1 × 10^−4^ M for priNH_2_H^+^Cl^−^, 1.2–2.4 × 10^−4^ M for secNHH^+^Cl^−^ and 2.1–5.2 × 10^−5^ M for terNH^+^Cl^−^. This decrease can be attributed to the formation of non-extractable HAuCl_4_ species in the aqueous phase as the HCl concentration in the solution increased [29].

In any case, and if the range of extractant concentrations used was compared, it can be clearly seen that there was an apparent gold(III) extractability order in the form: terNH^+^Cl^−^ > secNHH^+^Cl^−^ > priNH_2_H^+^Cl^−^, this order being based in the fact that with very diluted terNH^+^Cl^−^ concentrations, i.e., 2.1 × 10^−4^ M, gold(III) was extracted almost quantitatively from the aqueous solution, whereas in the case of priNH_2_H^+^Cl^−^, this extraction rate was achieved using a 0.2 M PPIL concentration that was a threefold concentration relationship, and the performance of secNHH^+^Cl^−^ was situated between both.

Experimental data were also numerically treated, and the results of these calculations are shown in Table 3.

It can be seen that the values of the extraction constants (K_ext_) increased with the increase of the HCl concentration in the aqueous phase and, thus, with the increase of the corresponding aqueous ionic strength (I_M_). It can also be noted that in the case of terNH^+^Cl^−^ PPIL, the values of K_ext_ were about two orders of magnitude greater than that corresponding to the other two PPILs, indicating the greater affinity of terNH^+^Cl^−^ to exchange its chloride anion with the AuCl_4_^−^ anion from the aqueous phase.

#### 2.2.4. Influence of the Initial Gold(III) Concentration in the Aqueous Phase on the Metal Extraction

The evaluation of this variable in the extraction of the precious metal was also under consideration, and organic phases containing different concentrations of the PPILs in toluene were mixed with aqueous solutions of gold (III) (0.005–0.05 g/L) in various HCl media. The results from these experiments are shown in Figure 2 for the three PPILs. Here, a general pattern can also be seen, since with the HCl concentration, a maximum of gold extraction was always reached in all the cases, though this maximum was wider as the initial gold concentration in the aqueous solution decreased. Further, the percentage of gold extraction at the various initial metal concentrations became more similar depending on the PPIL used to extract the gold.

Based on the values found in this and the previous subsection, one can ascertain that the extraction of gold(III) from HCl solutions by these PPILs responded to an anion exchange equilibrium:(5)$$\mathrm{terN}H^{+}Cl_{\mathrm{org}}^{-}+\mathrm{AuC}l_{4_{\mathrm{aq}}}^{-}\Leftrightarrow \mathrm{terN}H^{+}\mathrm{AuC}l_{4_{\mathrm{org}}}^{-}+Cl_{\mathrm{aq}}^{-}$$

The above is representative of terNH^+^Cl^−^ PPIL. In the other two cases, the equilibrium led to the formation of priNH_2_H^+^AuCl_4_^−^ or secNHH^+^AuCl_4_^−^ in the organic phase after the extraction process.

#### 2.2.5. Estimation of the (AuCl_4_^−^,H^+^) Interaction Coefficients

The different K_ext_ values with the ionic strength can be correlated by the Specific Interaction Theory [30], which can be used to estimate the interaction coefficient (ε) values of AuCl_4_^−^,H^+^ species in the aqueous solution.

Writing Equation (5) in a general form for the three PPILs, the extraction constant (K^0^) for the equilibrium was correlated with the ionic strength via the next expression:(6)$$K^{0}=K_{\mathrm{ext},m}\frac{\gamma {\left[\mathrm{PPIL}H^{+}\mathrm{AuC}l_{4}^{-}\right]}_{\mathrm{org}}\gamma {\left[Cl^{-}\right]}_{\mathrm{aq}}}{\gamma {\left[\mathrm{PPIL}H^{+}Cl^{-}\right]}_{\mathrm{org}}\gamma {\left[\mathrm{AuC}l_{4}^{-}\right]}_{\mathrm{aq}}}$$

In this expression, K_ext,m_ is the extraction constant in the molality scale, and γ is the activity coefficient of each species. If it was assumed that the organic phase had ideal behavior and input logarithms, the next equation resulted:(7)$$\log K^{0}=\log K_{\mathrm{ext},m}-\log\gamma_{{\left(\mathrm{AuC}l_{4}^{-}\right)}_{\mathrm{aq}}}+\log\gamma_{{\left(Cl^{-}\right)}_{\mathrm{aq}}}$$

The activity coefficient of an ion of charge z_i_ in a solution of a given ionic strength was defined as:(8)$$\log\gamma_{i}=-Z_{i}^{2}D_{I_{m}}+\Sigma \varepsilon_{\left(i,c\right)}I_{m}$$
where I_m_ represents the aqueous ionic strength in the molality scale; D_Im_ is the Debye-Hückel coefficient, also in the molality scale; and ε is the interaction coefficient between the pairs of charged species (AuCl_4_^−^ and Cl^−^ as anions and H^+^ as the cation) presented in this system.

Considering the above charged species and the ε term, the substitution in Equation (7) led to the expression:(9)$$\log K_{\mathrm{ext},m}=\log K^{0}+\left(\varepsilon_{\left(\mathrm{AuC}l_{4}^{-},H^{+}\right)}-\varepsilon_{\left(Cl^{-},H^{+}\right)}\right)I_{m}$$

Thus, plotting log K_ext,m_ versus I_m_ as a straight line of slope (ε_(AuCl4_^−^_,H_^+^_)_–ε_(Cl_^−^,_H_^+^_)_ and intercept log K^0^ might be obtained. The results from the various plots for the three PPILs used in this work are given in Table 4. The values of the interaction coefficients were calculated from the corresponding slope and considering that ε_(Cl_^−^_,H_^+^_)_ had a value of 0.12 [31,32].

#### 2.2.6. Gold Stripping from Metal-Loaded Organic Phases

The affinity of the gold(III) to be complexed by thiocyanate ions and to form the Au(SCN)_4_^−^ species (log β_4_ = 43.66) [33] in an aqueous medium was known. Thus, the stripping of gold from the various metal-loaded organic phases was investigated by the use of this complexing agent.

The results derived from the experiments carried out with PPIL organic phases loaded with 0.01 g/L Au(III) indicated that by using 0.75 M sodium thiocyanate solutions, the rate of gold recovered in the strip solution was 82%, 75% and 79% when using the organic phases of priNH_2_H^+^Cl^−^, secNHH^+^Cl^−^ or terNH^+^Cl^−^, respectively, after 15 min of contact between both phases at a temperature of 20 °C and using a V_aq_/V_org_ ratio of 1. A second equilibration of the gold’s partially depleted organic phase with fresh thiocyanate solution, under the same experimental conditions as above, allowed an overall gold strip rate of 99%.

#### 2.2.7. Precipitation of Zero-Valent Gold from Stripping Solutions

Due to the uses of gold nanoparticles in multiple fields, i.e., biological, chemical, medical, agricultural, etc. [34,35,36,37,38], it was of interest to consider gold recovery as a type of nanomaterial [39]; several methods were proposed [34,36,37], and one was the use of a reducing agent, such as sodium borohydride [40]. In the present work, under very gentle (50 min^−1^) stirring, 0.1 g of sodium borohydride was added to a thyocianate strip solution containing 5.1 × 10^−4^ M gold.

The reaction responsible for gold precipitation can be written as [29]:(10)$$\mathrm{Au}{\left(\mathrm{SC}N^{-}\right)}_{4}+\frac{1}{2}BH_{4}^{-}+2H_{2}O\to Au^{0}+3\mathrm{HSCN}+\frac{1}{2}B{\left(\mathrm{OH}\right)}_{4}^{-}+\mathrm{SC}N^{-}+\frac{1}{2}H_{2}$$

From the first addition of the reducing agent, a dark precipitate appeared, and at the end of the addition process, the precipitate was filtered and washed with distilled water. The as-obtained dry solid resulted in zero-valent gold nanoparticles (Figure 3) with some degree of agglomeration (Figure 4).

## 3. Materials and Methods

The amines Primene 81R (Rohm and Haas), Amberlite LA2 (Fluka) and Hostarex A327 (Sanofi) were used without further purification. All other chemicals were of AR grade.

The extraction and stripping experiments were done in thermostatted separatory funnels that provided mechanical shaking via four glass blade impellers.

In the case of the generation of the PPILs, equilibrated organic and aqueous phases were analyzed by titration with standard NaOH solutions and using blue bromothymol as the indicator. The results (Table 1) were given as the HCl distribution coefficient (D_HCl_), defined as:(11)$$D_{\mathrm{HCl}}=\frac{{\left[\mathrm{HCl}\right]}_{\mathrm{org}}}{{\left[\mathrm{HCl}\right]}_{\mathrm{aq}}}$$

Gold in the aqueous phases was analyzed by AAS (Perkin Elmer 110B spectrophotometer), and the corresponding concentration in the organic phase ([Au]_0_) was calculated by the mass balance, and the percentage of gold extracted in the organic phase was calculated as:(12)$$\%\mathrm{gold}=\frac{{\left[\mathrm{Au}\right]}_{aq,0}-{\left[\mathrm{Au}\right]}_{\mathrm{org}}}{{\left[\mathrm{Au}\right]}_{aq,0}}\times 100$$
where [Au]_aq,0_ is the initial gold concentration in the aqueous phase. The percentage of gold stripped, [Au]_st_, was calculated as:(13)$$\%\mathrm{gold}=\frac{{\left[\mathrm{Au}\right]}_{\mathrm{st}}}{{\left[\mathrm{Au}\right]}_{\mathrm{org},0}}\times 100$$
where [Au]_org,0_ is the gold concentration in the metal-loaded organic phase.

The TEM image was obtained at 120 KV using JEOL JEM 1400 equipment.

## 4. Conclusions

The results derived from this investigation indicated the usefulness of quaternary ammonium chloride PPILs to extract gold(III) from HCl media, though the performance demonstrated by the PPIL derived from a tertiary amine was much better than the ones by PPILs derived from secondary and primary amines.

In all the cases, equilibrium was reached at a short time, and the three systems showed the same behavior with respect to temperature: exothermic and spontaneous extraction processes, with a decrease in randomness as a consequence of gold uploading in the respective organic phase.

Further, in all three extraction systems, the extraction constant depended on the aqueous ionic strength, and on this basis, the interaction coefficients of the AuCl_4_^−^,H^+^ pair were estimated.

The sodium thiocyanate medium appeared to respond well as a gold stripper from the metal-loaded organic phases, and from these stripped phases, gold could be conveniently precipitated as zero-valent gold nanoparticles (partially aggregated).

## Figures and Tables

**Figure 1 ijms-24-06305-f001:**
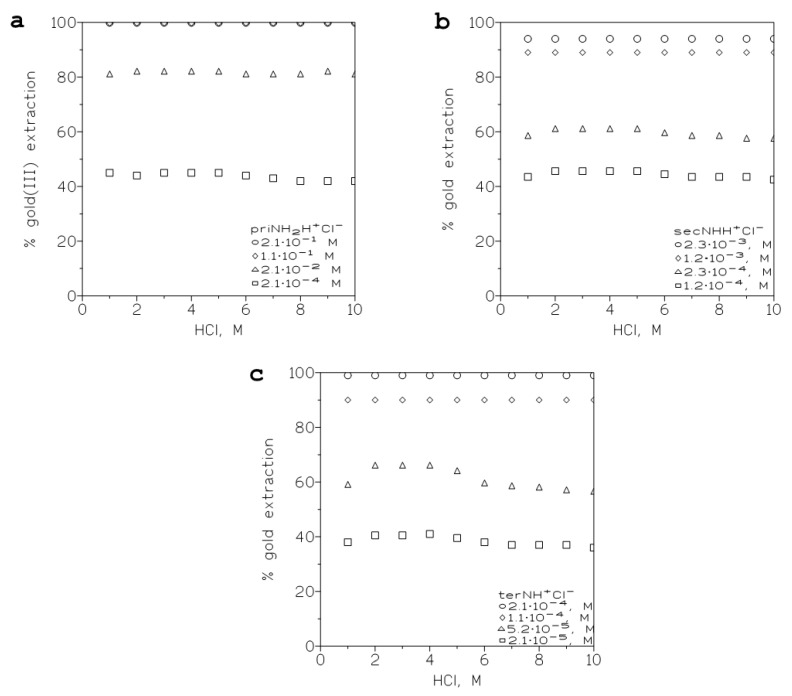
Variation of the percentage of gold(III) versus the HCl concentration in the aqueous phase at various PPILs concentrations in toluene. (**a**): priNH_2_H^+^Cl^−^. (**b**): secNHH^+^Cl^−^. (**c**): terNH^+^Cl. Equilibration time: 5 min. Temperature: 20 °C.

**Figure 2 ijms-24-06305-f002:**
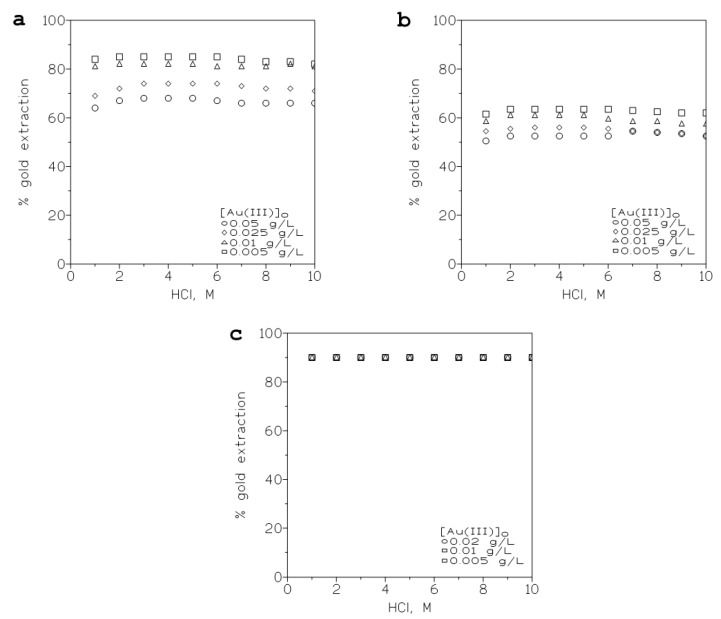
Variation of the percentage of gold(III) extraction versus HCl concentration in the aqueous phase at various initial metal concentrations. (**a**): 2.1 × 10^−2^ M priNH_2_H^+^Cl^−^. (**b**): 2.3 × 10^−4^ M secNHH^+^Cl^−^. (**c**): 1.1 × 10^−4^ M terNH^+^Cl^−^. Equilibration time: 10 min. Temperature: 20 °C.

**Figure 3 ijms-24-06305-f003:**
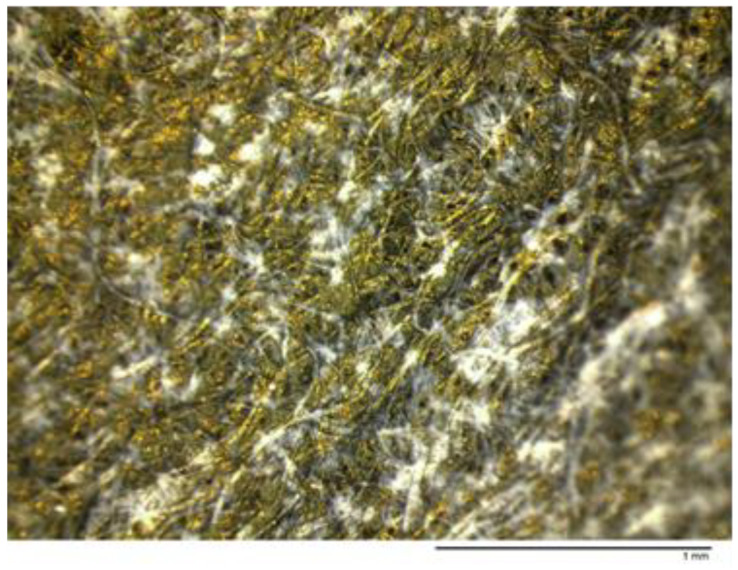
Zero-valent gold.

**Figure 4 ijms-24-06305-f004:**
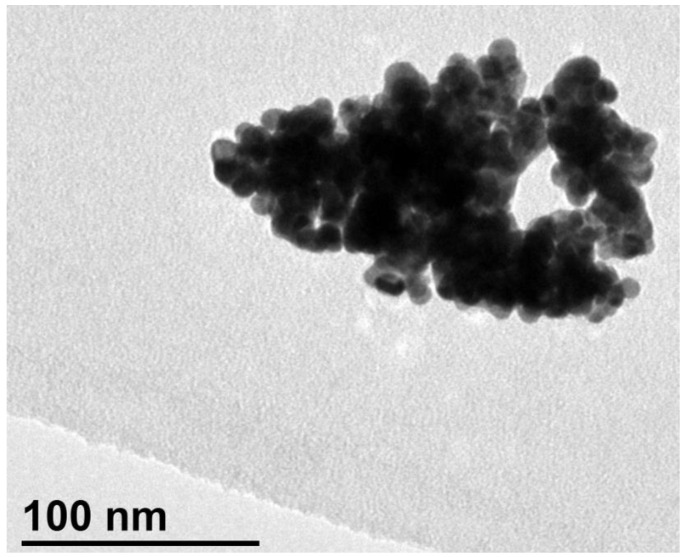
TEM image of the precipitated zero-valent gold.

**Table 1 ijms-24-06305-t001:** Generation of the PPILs.

Precursor	Concentration, M	D_HCl_	PPIL Formed	Acronysm	log K_ext_
Primene 81R	0.11	0.11	RNH_2_H^+^Cl^−^	priNH_2_H^+^Cl^−^	2.99 ^a^
	0.21	0.24			
	0.41	0.65			
	0.82	3.6			
Amberlite LA2	0.06	0.063	R´R´´NHH^+^Cl^−^	secNHH^+^Cl^−^	2.90 ^b^
	0.12	0.13			
	0.24	0.31			
	0.48	0.89			
Hostarex A327	0.06	0.056	R_3_NH^+^Cl^−^	terNH^+^Cl^−^	2.65 ^c^
	0.12	0.12			
	0.21	0.26			
	0.42	0.72			

^a^ U = 3.1 × 10^−4^, ^b^ U = 3.0 × 10^−4^, ^c^ U = 2.3 × 10^−5^.

**Table 2 ijms-24-06305-t002:** Influence of the temperature on gold extraction.

PPIL, M	1000/T, K^−1^	log D_Au_	ΔH°, kJ/mol	ΔS°, J/mol K	ΔG°, kJ/mol
^a^ 2.1 ××10^−2^	3.4	^a^ 1.6	−10	−31	−1
	3.3	1.5			
	3.2	1.2			
	3.1	1.1			
	3.0	1.0			
^b^ 2.3 × 10^−4^	3.4	^b^ 4.6	−21	−59	−4
	3.3	3.6			
	3.2	2.6			
	3.1	2.0			
	3.0	1.7			
^c^ 5.3 × 10^−5^	3.4	^c^ 0.28	−26	−82	−2
	3.3	0.15			
	3.2	0.04			
	3.1	−0.17			
	3.0	−0.25			

^a^ priNH_2_H^+^Cl^−^ (r^2^ = 0.979), ^b^ secNHH^+^Cl^−^ (r^2^ = 0.996), ^c^ terNH^+^Cl^−^ (r^2^ = 0.983). V_org_/V_aq_ = 1. Equilibration time: 5 min.

**Table 3 ijms-24-06305-t003:** Values of K_ext_ at various HCl concentrations.

HCl, M	I_M_	priNH_2_H^+^Cl^−^	secNHH^+^Cl^−^	terNH^+^Cl^−^
1	1.02	4.9 × 103	7.1 × 103	6.0 × 105
2	2.02	8.3 × 103	1.4 × 104	7.6 × 105
3	3.19	1.3 × 104	2.2 × 104	1.1 × 106
4	4.36	1.7 × 104	3.0 × 104	1.6 × 106
5	5.57	2.2 × 104	3.9 × 104	2.0 × 106
6	6.85	2.8 × 104	4.9 × 104	2.5 × 106
7	8.19	3.2 × 104	5.6 × 104	3.9 × 106
8	9.61	3.7 × 104	6.6 × 104	4.9 × 106
9	11.11	4.2 × 104	7.6 × 104	6.0 × 106
10	12.69	4.7 × 104	8.7 × 104	7.1 × 106

I_M_ = ionic strength in the molar scale. K_ext_ values rounded to one decimal.

**Table 4 ijms-24-06305-t004:** Results of plotting of log K_ext,m_ versus I_m_.

I_m_	^a^ log K_ext,m_	^b^ log K_ext,m_	^c^ log K_ext,m_
1.02	3.69	3.85	5.78
2.02	3.92	4.16	5.88
3.19	4.10	4.34	6.06
4.36	4.24	4.48	6.20
5.57	4.35	4.59	6.31
6.85	4.44	4.69	6.40
8.19	4.50	4.75	6.59
9.61	4.57	4.82	6.69
11.11	4.62	4.88	6.78
12.69	4.67	4.94	6.85
log K0	3.80	4.01	5.74
ε(AuCl4−,H+)	0.20	0.20	0.21
r2	0.906	0.886	0.979

^a^ Values for priNH_2_H^+^Cl^−^. ^b^ Values for secNHH^+^Cl^−^. ^c^ Values for terNH^+^Cl^−^.

## Data Availability

Not applicable.

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
