# Peer review of "On the Use of Pseudo-Protic Ionic Liquids to Extract Gold(III) from HCl Solutions"

_ijms, 2023, doi:10.3390/ijms24076305_

Round 1

Reviewer 1 Report

The Authors present an interesting discussion and modeling of gold extraction by amine-based ionic liquids. The works shows a complete investigation of the effect of experimental parameters on the efficiency of the extraction, release (by stripping of metal-loaded organic phase) and final valorization with borohydride precipitation as gold nanoparticles). The modeling is meaningful and the work offers basic and fundamental information that may be of interest for researchers involved in metal recovery and valorization. This is a very promising and the reader is logically waiting the continuation of this work in terms of application to more complex effluents (containing competitor or co-metals) up to the application to industrial leachates. The main question the reader may have at this stage is … how the type of amine used for the synthesis of PPILs may affect the selectivity of the extraction.

Editing

Few typing mistakes (L34? L80; L109, L133; L142; L144; L147, L238)

Editing equations (re-arrange numbering)

Complete caption Figure 2c (missing terNH+Cl-)

Table 4: Please, identify each column with the relevant PPILs name.

Specific questions:

(a) Is there any explanation to the faster extraction for sec/ter PPILs (compared with pri-PPIL)?

(b) Do the authors have tentative explanation for the effect of the nature of the amine on the thermodynamic properties?

(c) As far as we can read in Section 2.2.6, gold stripping efficiency is not affected by the type of PPIL. Is there any reason to explain that despite the stronger interaction of gold tetrachloride anions with ter-NH+Cl-, the levels of release are exactly the same?

(d) Is there any possibility to qualify the gold NPs (by TEM?) obtained after precipitation as zero valent gold (with sodium borohydride)? Indeed, the interest of gold NPs for the different expected uses is strongly impacted by the size of nano-objects.

Reviewer 2 Report

The manuscript deals with the extraction of Au and presented some interesting work. Nevertheless, a design of experiments (DoE) to optimize the procedure and find suitable reaction conditions is missed.

Some further comments should be answered:

Abstact:

Please explain pri-NH2H+Cl- (derived from 9 the primary amine Primene 81R), sec-NHH+Cl- (derived from the secondary amine Amberlite LA2) 10 and ter-NH+Cl- (derived from the tertiary amine Hostarex A327).

page 2 line 47:

under debate [19,20]. Why? Please give examples of critical IL's.

page 3 line 59:

PILLS, security or safety?, Whose regulations? For what?

page 3, line 76, delete amine in ammonium amine salt

page 3, line 77, delete acid in acidity or acid

page 3 line 80-81, please explane nature of these commercial materials

page 4, line 102, why this concentration?

page 8, line 251, meanwhile Dupont

Reviewer 3 Report

In last decades there are many investigations about recovery of different very useful but rare and/or expensive metals. In this manuscript (Communication) authors, F. J. Alguacil and J. I. Robla, represented their investigation about extraction of gold from acid media using pseudo-protic ionic liquids generated from the reaction of a primary, secondary and tertiary amine with HCl. This scientific paper is a continuation of the work of the author F. J. Alguacil regarding investigation of recovery of gold (and some other metals) under different conditions (different acids, concentrations of acid and metal, ionic liquids, temperature, reaction time, etc.). The focus here is on the use of pseudo-protic ionic liquids.

They explained their conclusions, based on the obtained results, very well so I recommend this paper for publication in special Issue: Advances in Ionic Liquids and Their Various Applications, but there are a few suggestions and things that need to be corrected:

Page 1, lines 28,29 - It would be useful to write a reference/s for the listed method/techniques.

Page 2, lines 52,53 - sentence should be corrected, it is a little bit unclear (in line 53)

Page 3, line 109 – “o” in the title

Page 4, line 133 - “imvestigate” should be corrected

Page 4, line 142 – is there any conclusion why “progressive decrease in the extraction occurred from 6M HCL concentration”? If yes, it would be useful to write that here.

About that progressive decrease in the extraction, that is not so visible in the picture 1, but it was written: “this decrease was only observed with the lower extractant concentrations”, it would be useful to write the concentration of extractant at which that progressive decrease was noticed.

Page 4, line 147 – “doluted” should be corrected

Page 5, Figure 1. -  In Figure 1a, there are only 3 concentrations. If 2.1E-1 and 1.1E-1 are the same, because of safe of clarity it would be good to make that symbols more visible.

Round 2

Reviewer 2 Report

now the paper is fine